# Implementing a smoking cessation intervention for people experiencing homelessness: Participants' preferences, feedback, and satisfaction with the 'power to quit' program

Oluwakemi Ololade Odukoya[1]*, Ekland A. Abdiwahab[2], Tope Olubodun[3], Sunday Azagba[4], Folasade Tolulope Ogunsola[5], Kolawole S. Okuyemi[4]

1 Department of Community Health and Primary Care, College of Medicine, University of Lagos, Lagos State, Nigeria, 2 Department of Epidemiology and Biostatistics, University of California San Francisco, San Francisco, California, United States of America, 3 Department of Community Health and Primary Care, Lagos University Teaching Hospital, Lagos, Lagos State, Nigeria, 4 Department of Family & Preventive Medicine, University of Utah, Salt Lake City, Utah, United States of America, 5 Department of Medical Microbiology and Parasitology, College of Medicine, University of Lagos, Nigeria

☯ These authors contributed equally to this work.
* drolukemiodukoya@yahoo.com

**Data Availability Statement:** The data files and data dictionary are available from the ICPSR

## Abstract

### Background

Smoking rates among populations experiencing homelessness are three times higher than in the general population. Developing smoking cessation interventions for people experiencing homelessness is often challenging. Understanding participant perceptions of such interventions may provide valuable insights for intervention development and implementation. We assessed participants' satisfaction and preferences for the Power to Quit (PTQ) program.

### Methods

PTQ was a 26-week community-based smoking-cessation RCT among people experiencing homelessness. A total of 315 of the 430 enrolled participants completed the 26 week-study feedback survey. Overall program satisfaction was measured on a 5-point Likert scale by asking the question "Overall, how satisfied were you with the Power to Quit Program?" Analyses were conducted to identify factors associated with overall program satisfaction.

### Results

Participants were mostly male (74.9%), African American (59.0%), 40 years and older (78.2%), and not married or living with a partner (94.9%). Visa gift cards were the most preferred incentive followed by bus tokens and Subway restaurant coupons. The patch and counseling were the top-ranked intervention component, 55.3% rated the patch as very helpful; 59.4% felt counseling sessions was very helpful; 48.6% found reminder phone calls

database. Accessible at https://www.openicpsr.org/openicpsr/project/165321/version/V1/view.

**Funding:** Research reported in this publication was supported by the National Heart Lung, and Blood Institute of the National Institutes of Health (https://www.nih.gov/) under Award Number R01HL081522. KSO received the award. The protected time for the contribution of OO (Oluwakemi Odukoya) towards the research reported in this publication was supported by the Fogarty International Center of the National Institutes of Health under the Award Number K43TW010704. The content is solely the responsibility of the authors and does not necessarily represent the official views of the National Institutes of Health. The funders had no role in study design, data collection and analysis, decision to publish, or preparation of the manuscript.

**Competing interests:** The authors have declared that no competing interests exist.

or messages most helpful for appointment reminders. Majority (78.7%) said they were very satisfied overall, 80.0% were very satisfied with the program schedule, and 85.4% were very satisfied with program staff. Race and age at smoking initiation were predictors of overall program satisfaction. African American/Black participants were 1.9 times more likely to be satisfied with the program compared to White participants.

## Conclusion

Majority of the participants of PTQ were satisfied with the program. This study supports the acceptability of a smoking cessation program implemented in a population experiencing homelessness. The high rate of satisfaction among African American participants may be in part because of race concordance between participants, study staff, and community advisory board. Including staff that have a shared lived experience with participants in a smoking cessation study may improve the participant satisfaction within such studies.

## Introduction

Smoking remains a leading preventable cause of morbidity and mortality globally and in the United States [1]. Smoking is particularly high among the approximately 610,000 individuals experiencing homelessness in the U.S [2–5]. About 73%-80% of individuals experiencing homelessness are smokers compared to 14% of the general population [6–8]. Tobacco smoking increases the risk for obstructive lung disease, cardiovascular disease, cancer, and other chronic conditions [9, 10]. Homelessness exacerbates these risks and reinforces the critical need for smoking cessation interventions in populations experiencing homelessness [11–16]. However, there is a concern that the design and implementation of effective interventions may be hampered by competing priorities that occur commonly among smokers experiencing homelessness. For example, the need for mental health services, drug and alcohol services, food and shelter, and the transient nature of this population [17–19].

Participant satisfaction has long been recognized as an important element of health services research, and a report by the Institute of Medicine (National Academy of Medicine) identified satisfaction as a key indicator of quality of care [20]. Studies have found an association between participant satisfaction and program-related outcomes. In a study of Korean smokers, participants who were satisfied with Quitline services were more likely to maintain cessation for up to a year compared to those who were not satisfied (14.7% vs. 2.8%). Even among participants who relapsed, those who were satisfied with the intervention reduced the number of cigarettes smoked daily compared to those who were not (i.e., 37.6% versus 18.4%). In addition, participants who were satisfied with the contents of counseling and coaching protocols were more likely to quit smoking compared to those who were dissatisfied [21].

In a longitudinal study of 502 adults who began treatment for substance use, Carlson and Gabriel (2001) found that individuals who reported high levels of satisfaction with program services were two times more likely to abstain from substance use than those who reported a low level of satisfaction [22]. Others have also found positive associations between participant satisfaction and use of program services, longer treatment retention, and completion of program [23, 24]. These findings suggest that understanding participant satisfaction may not only be important for short-term outcomes such as retention and completion but also have implications for long-term outcomes such as cessation and/or relapse.

Few studies have examined participant satisfaction with smoking cessation programs among populations experiencing homelessness. This paper uses data from the Power to Quit study to assess participants' preferences for various aspects of the smoking cessation program, their opinions of the main components of the program, and their satisfaction with program components. In addition, we identify factors associated with greater program satisfaction among smokers experiencing homelessness. Results from this study may be helpful to others designing smoking cessation programs for populations experiencing homelessness.

## Materials and methods

### The power to quit program

The Power to Quit program (Trial Registration Number: NCT00786149) was a randomized-controlled trial of smokers experiencing homelessness in Minneapolis and St. Paul, Minnesota. A total of 430 participants were randomized into two groups. At baseline, both arms received a two-week supply of 21-mg NRT patches. Over an eight-week period, both arms received an additional two-week supply of the nicotine patch every two weeks. In addition, the intervention group received six 15–20-minute participant-led counseling sessions (motivational interviewing) while the control group received standard care (one brief counselor-led session lasting 10–15 minutes). The primary aim of the intervention was to evaluate the efficacy of motivational interviewing (MI) as well as nicotine replacement therapy (NRT) in smokers experiencing homelessness [25]. Our study assessed participants' satisfaction with the patch and counseling sessions for both groups.

The study lasted 26 weeks and in total there were 15 counseling and retention visits. At each visit, participants received incentives as compensation for their time and effort. At longer visits that included surveys, participants received $20 gift cards and two bus tokens ($3 value). For attending brief retention visits and the week 8 end of treatment visit, participants received $10 gift cards, two bus tokens, and another small gift item. Small gift items included playing cards, tote bags, movie passes, water bottles, T-shirts, and personal care items (e.g., soap, toothbrush, washcloth). For attending the final 6-month visit, participants received a $40 gift card and a sweatshirt. For participants who attended all 15 sessions, the monetary incentives totaled $275 over the 6-month study period.

### Participants

Eligible participants were recruited over 15 months (May 2009 to August 2010) from eight emergency homeless shelters and transitional homes in the Twin Cities (Minneapolis and St. Paul, Minnesota). Study sites were located in the downtown/city center easily accessible by public transportation such as city-operated buses and light rail. Researchers recruited participants by conducting health fairs, holding informational interviews, posting flyers, and announcements at homeless shelters. Participants also helped to recruit by word of mouth. Inclusion criteria was a confirmation of homelessness as defined by the Stewart B. McKinney Act [26] as 'any individual who lacks a fixed, regular and adequate nighttime residence'; or 'one whose primary nighttime residence is a supervised publicly or privately-operated shelter designed to provide temporary living accommodations, transitional housing, other supportive housing program or a public or private place not meant for human habitation.' Participants were also classified as experiencing homelessness if they were without a home and had been staying with family or friends for up to three months [26, 27]. Individuals were also included if they 1) were a current smoker who had smoked at least 100 cigarettes in their entire lifetime; 2) reported smoking at least 1 cigarette every day over the past 7 days and CO score above 5; 3) ≥ 18 years of age; 4) lived in the Twin Cities for at least 6 months and planned to stay for the

following 6 months of the study; 5) wished to use the nicotine patch for 8 weeks and 6) were willing to complete 15 total appointments over the 26-week study period. Participants were excluded if they 1) used a tobacco quit aid in the previous 30 days; 2) had a cognitive impairment; 3) had suicidal ideation in the last 14 days, or 4) had a major medical condition within the previous 30 days. Participants were also excluded if they scored greater than five on items assessing psychotic symptoms. Full details of the study design have been published [27].

## Data collection

Validated questionnaires were used to collect survey data. Metric measurements of height and weight were collected to calculate body mass index. Carbon monoxide (CO) and saliva cotinine were assessed as biomarkers of tobacco use. The participants were asked to exhale into a carbon monoxide monitor. Patch adherence was measured by: 'patch checks' (visual verification of whether a participant was wearing a patch); 'patch counts' (documenting the number of patches left in the participant's possession); and administration of the Morisky scale, a self-reported adherence scale modified to assess adherence to NRT patch.

Study data was collected at baseline, week 1, week 2, week 4, week 6, week 8, week 10, week 12, week 14, week 16, week 18, week 20, week 24, and week 26. Because of the nature of the study population, timings for assessments and counselling sessions were flexible.

Data analyzed in this study was limited to the baseline and week 26 feedback survey. At the baseline survey, information collected included: demographic characteristics, housing, general health status, smoking history, quitting history, cigarette accessibility, nicotine dependence, confidence in quitting smoking, self-efficacy, mental health measures including depression and perceived stress, alcohol use, drug abuse/dependence, exhaled carbon monoxide, height, weight, and cotinine. At the 26-week feedback survey, information obtained included: data on preferred program components and incentives, adherence to nicotine replacement, difficulties experienced with using the patch, reasons for missing appointments, what helped the most as appointment reminders, motivations for keeping appointments, helpfulness of counseling sessions and program satisfaction. Outcome data such as smoking characteristics, exhaled CO, weight, and cotinine were repeated at the 26-week survey. Some of the information collected in weeks 2, 4, 6, and 8 surveys included information on housing, adverse effects, tobacco cessation, patch adherence, confidence in quitting smoking and exhaled CO.

## Measures

**Outcome variable.** Overall program satisfaction was our key dependent variable. This was measured on a 5-point Likert scale. Participant satisfaction was measured by asking: "Overall, how satisfied were you with the Power to Quit Program?" The response options were: 1)Not satisfied at all; 2) Somewhat unsatisfied; 3) Neutral; 4) Somewhat satisfied; 5) Very satisfied. In univariate and multivariable analysis, the outcome variable overall satisfaction was dichotomized and given a value of 1 if "Very satisfied" and a value of 0 for all other responses.

**Exposure variables.** The main components of the intervention were the nicotine patch and counseling sessions. Incentives for participation included Visa gift cards, bus tokens, calendar/organizers, tote bags, restaurant coupons, movie passes, polo shirts, water bottles, soap and washcloths, back massagers, and a dental package. Preferences for program components were assessed by asking participants: "Which items were the most helpful to you?" Participants were asked to number program components on a scale of 1 to 3 in the order that they were most helpful -1(most helpful), 2(helpful), 3(least helpful). The response options were: 1) Reading materials; 2) Individual counseling sessions; 3) Nicotine patch; 4) Community mobilizer contacts; 8)Don't know/don't remember. Community mobilizers are research assistants on the

study team who were either homeless at the time of the study or had recently experienced homelessness. Similarly, preferences for program incentives were assessed by asking participants; "Choose three of the items that you liked the most and number them from 1 to 3 in the order of preference -1(liked the most), 2(liked), 3(liked the least). The response options were:1) Bus tokens; 2) Tote bag; 3) Calendar/organizer; 4) Visa gift cards; 5) Restaurant coupons (Subway); 6) Movie passes; 7) Polo shirt; 8) Water bottle; 9) Soap and washcloth; 10) Back massager; 11) Dental package; 88)Don't know/don't remember. The first ranked choice of each respondent was used for our analysis. The "Don't know/don't remember" responses were dropped from the analysis.

**Socio-demographic covariates.** Respondents ages were recoded into five groups i.e., < 30 years, 30–39 years, 40–49 years, 50–59 years, ≥60 years. Race/Ethnicity was recoded as African American/Black, White, and Other. Marital Status was recoded as married/living with significant other, divorced/widowed/separated, and never been married. Education was recoded as less than high school and at least high school education. Employment was recoded as currently employed and currently unemployed. Monthly income was recoded as <$400, ≥$400.

**Health and psychosocial covariates.** Self-reported general health was assessed by asking: "In general, would you say your health is?" The response options were: 1) Excellent; 2) Very good; 3) Good; 4) Fair; 5) Poor. Depression was assessed with PHQ-9. Respondents were asked how often they have been bothered by a range of nine problems over the preceding 2 weeks. Response options for each question were: 0) Not at all; 1) Several days; 2) More than half the days; 3) Nearly every day. Responses were summarized and respondents with scores ranging from 0–4 were categorized as having no depression, scores of 5–9 were categorized as mild depression, scores of 10–19 were categorized as moderate depression, and scores of 20–27 were categorized as severe depression [28]. Stress was assessed using the perceived stress scale. Respondents were asked four questions to assess how often they experienced stress in their life in the past 30 days. Response options were: 0) Never; 1) Rarely, 2) Sometimes; 3) Often; 4) Very often. Responses were summarized and respondents with scores ranging from 0–5 were categorized as low perceived stress, scores from 6–10 were categorized as moderate perceived stress, and scores from 11–16 were categorized as high perceived stress [29, 30].

**Homelessness covariates.** The number of times participants experienced homelessness in the past 3 years was assessed by asking: "During the last three years, how many separate times have you been homeless or without a regular place to live?" Responses were recoded as once, twice, and three or more times.

**Smoking-related covariates.** Age at smoking initiation was grouped as ≤10 years, 11–20 years, 21–30 years, 31–40 years, and ≥41 years. Time to first cigarette was assessed by asking respondents: "How soon after you wake up do you smoke your first cigarette?" Responses were dichotomized as ≤5 minutes and >5 minutes. The number of past-year 24-hour quit attempts was assessed by asking: "In the last year, on how many times have you seriously tried to quit smoking for at least 24 hours?" This was recoded as 1–10 times, 11–20 times, 21–30 times, 31–40 times. Confidence to quit smoking was assessed by asking a series of numerically scored questions with higher responses indicating higher levels of confidence. Scores were reported in means and standard deviation.

**Substance use and dependence covariates.** Drug and alcohol dependence was assessed using the 3-item Rost-Burnam screener. Drug dependence was measured by asking participants: "Have you ever used one of these drugs on your own more than 5 times in your lifetime?", "Did you ever find you needed larger amounts of these drugs to get an effect or that you could no longer get high on the amount you used to use?", "Did you ever have emotional or psychological problems from using drugs-like feeling crazy or paranoid or depressed or uninterested in things?" Response options were: 0) No; 1) Yes. Scores were summed and a

score of 0 was categorized as not dependent and a score of ≥1 was categorized as dependent. Alcohol dependence was measured by asking participants: "Did you ever think that you were an excessive drinker?", "Have you ever drunk as much as a fifth of liquor in one day?", "Has there been a period of two weeks when every day you were drinking 7 or more beers, 7 or more drinks or 7 or more glasses of wine?". Response options were: 0) No; 1) Yes. Scores were summed and a score of 0 was categorized as not dependent and a score of ≥1 was categorized as dependent.

## Statistical analysis

Pearson's chi-square test and Fisher's exact test (as appropriate) were used to examine bivariate associations between study variables. Logistic regression analyses were used to examine the associations between overall satisfaction and program components and incentives. In multivariable analysis, variables significant at p<0.10 in the bivariate analysis were imputed in the logistic regression in order to include variables tending towards a positive significance [31]. In addition, we controlled for age, race/ethnicity, and gender as previous studies have shown that greater program satisfaction is associated with demographic characteristics [32, 33]. All of the statistical analyses were performed using Stata 16.0 and p-values <0.05 were considered statistically significant at multivariable analysis [34].

## Ethical considerations

Study procedures were approved and monitored by the Institutional Review Board of the University of Minnesota Medical School (Study Number: 1307M39761). Written informed consent was obtained from participants prior to data collection.

## Results

### Sociodemographic, health, and substance use characteristics

Of the 430 participants enrolled in the study, 315 (73.3%) completed the week-26 feedback survey of which the majority were male (74.9%), African American/Black (59%), and 40 years or older (78.2%). More than half of the respondents (51.8%) were never married and 76.5% were high school graduates or had a General Educational Development (GED) qualification. Majority were unemployed (90.2%) and most (67.9%) had a monthly income of less than $400. Regarding their health, about a third (32.3%) reported having good health and an over half (57.2%) reported no depression or mild depression. Most respondents (77.4%) started smoking regularly between the ages of 11 and 20. For almost half of the respondents (46.4%), time to first cigarette was ≤5 minutes and many respondents (55.5%) had at least one unsuccessful 24-hour quit attempt in the past year. On a scale of 0 (not confident) to 10(extremely confident), the mean confidence to quit was 7.30 (2.41). About 10% of the respondents abstained from cigarette at the end of the study. More than half of the respondents (59.4%) had a positive screen for alcohol dependence, and majority (84.1%) had a positive screen for drug dependence (Table 1).

### Preferred program components and program incentives

The patch was the top-ranked component followed by counseling sessions (Fig 1).

Of the incentives offered to participants for participation, Visa gift cards were the most preferred incentive followed by bus tokens and Subway restaurant coupons (Fig 2).

**Table 1. Respondents' demographic, health, psychosocial and smoking related characteristics.**

| | Frequency | Percentage |
|---|---|---|
| *Demographic characteristics* | | |
| **Age (mean (SD)) (n = 314)** | 45.7(9.8) | |
| **Gender (n = 315)** | | |
| Male | 237 | 75.5 |
| Female | 78 | 24.8 |
| **Race/Ethnicity (n = 315)** | | |
| White | 106 | 33.7 |
| African American/Black | 186 | 59.0 |
| Other# | 23 | 7.3 |
| **Marital Status (n = 313)** | | |
| Married/Living with significant other | 16 | 5.0 |
| Divorced/Widowed/Separated | 135 | 43.1 |
| Never been married | 162 | 51.8 |
| **Highest Level of Education (n = 315)** | | |
| < High school | 74 | 23.5 |
| ≥ High school | 241 | 76.5 |
| **Employment (n = 315)** | | |
| Currently employed | 31 | 9.8 |
| Currently unemployed | 284 | 90.2 |
| **Monthly income (n = 315)** | | |
| <$400 | 214 | 67.9 |
| ≥$400 | 101 | 32.1 |
| *Homelessness characteristic* | | |
| **Number of times homeless in past 3 years (n = 313)** | | |
| Once | 124 | 39.6 |
| Twice | 83 | 26.5 |
| Thrice or more | 106 | 33.9 |
| *Health and psychosocial characteristics* | | |
| **Self-reported general health (n = 313)** | | |
| Excellent | 48 | 15.3 |
| Very Good | 89 | 28.4 |
| Good | 101 | 32.3 |
| Fair | 61 | 19.5 |
| Poor | 14 | 4.5 |
| **Depression PHQ9 ≥10 (n = 313)** | | |
| None | 97 | 31.0 |
| Mild | 82 | 26.2 |
| Moderate | 63 | 20.1 |
| Moderately severe | 50 | 16.0 |
| Severe | 21 | 6.7 |
| Mean (SD) | 8.97 (6.54) | |
| **Stress PSS4, past 30 days (n = 313)** | | |
| Low stress | 34 | 10.9 |
| Moderate stress | 235 | 75.1 |
| High stress | 44 | 14.0 |
| Mean (SD) | 2.03 (0.5) | |
| *Smoking-related characteristics* | | |

(*Continued*)

**Table 1.** (Continued)

|  | Frequency | Percentage |
|---|---|---|
| **Age started smoking regularly (n = 314)** |  |  |
| ≤ 10 years | 26 | 8.3 |
| 11–20 years | 243 | 77.4 |
| 21–30 years | 33 | 10.5 |
| 31–40 years | 7 | 2.2 |
| ≥ 41 years | 5 | 1.6 |
| Mean (SD) | 16.6 (6.2) |  |
| **Time to first cigarette (n = 315)** |  |  |
| ≤5 minutes | 146 |  |
| >5 minutes | 169 | 53.6 |
| **Number of 24-hour quit attempts past year (n = 310)** |  |  |
| 0 times | 126 | 40.7 |
| 1–10 times | 172 | 55.5 |
| 11–20 times | 7 | 55.5 |
| >20 times | 5 | 2.3 |
| mean (SD) | 2.5 (4.9) |  |
| **Confidence to quit, (mean(SD)) (n = 315)** | 7.3 (2.4) |  |
| *Substance abuse variables* |  |  |
| **Had a positive screen for alcohol dependence (n = 313)** |  |  |
| No | 50 | 40.6 |
| Yes | 186 | 59.4 |
| **Had a positive screen for drug dependence (n = 314)** |  |  |
| No | 50 | 15.9 |
| Yes | 264 | 84.1 |

#Other categories include America Indian/Alaskan Native, Hispanic, Filipino, Jewish, Indian, Irish, Multi-racial

## Participants' feedback on the program components

Table 2 shows the participants feedback on the program components. Over half of the respondents (55.3%), rated the patch as very helpful. Of those that encountered some difficulties using the patch, the most common reason was remembering to put it on every day (26.6%). Other challenges included burning/itching side effect (21.0%) and difficulty sleeping (16.1%). Although less than half of the respondents continued the patch after the study ended (45.7%), 81.6% said they would have continued if provided. Similarly, more than half of the participants (59.4%) felt the counseling sessions were very helpful. Among those that missed appointments, the most common reason for missing appointments was that they forgot (39.7%). Reminder phone calls or messages (48.6%) and the desire to quit smoking (43.6%) motivated most participants to keep their appointments. Furthermore, nearly two-thirds (63.9%) of participants indicated they would like to continue with counseling.

## Program satisfaction

Majority of participants (78.7%) said they were very satisfied with the overall program Majority also said they were very satisfied with the location (81.3%) and with the counselors (86.4%). Most of the respondents (80.0%) were very satisfied with the program schedule, i.e., appointment times, frequency, and length of sessions. Majority (85.4%) were very satisfied with the

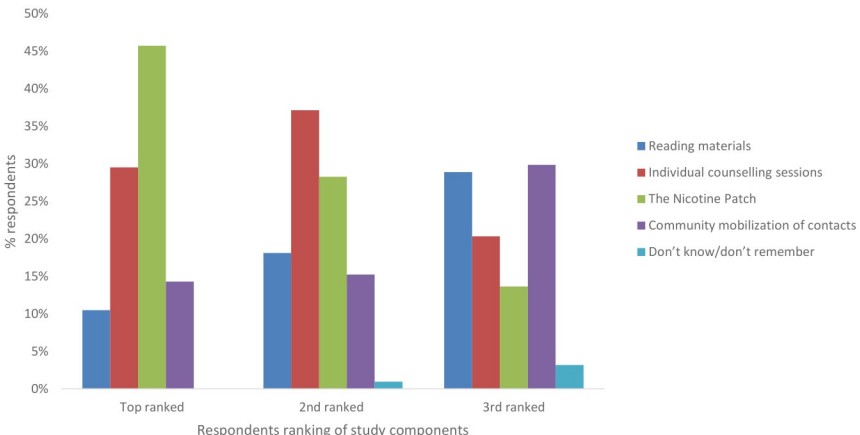

**Fig 1. Participants preference for study components.**

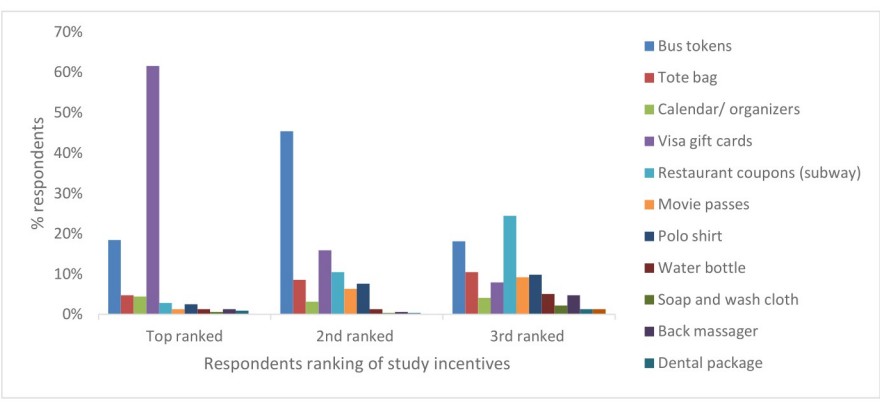

**Fig 2. Participants preference for study incentives.**

program staff. On a scale of 0(least) to 5(most), 4.9(0.4) reported feelings of being treated with respect and feeling that the research was adequately explained (Table 3).

## Predictors of program satisfaction

Race/ethnicity, age started smoking and confidence to quit showed statistically significant association with participants satisfaction at p<0.10 (Table 4). Race was a predictor of overall programme satisfaction. African American/Blacks were 1.84 times more likely to be very satisfied with the program compared with White or other participants. Age at smoking initiation was also a predictor of participants satisfaction; participants who started smoking at a younger age were more likely to be very satisfied with the program (Table 5).

## Discussion

This paper evaluates participants' preferences and satisfaction with the components of a smoking cessation intervention among smokers experiencing homelessness. While participants were very satisfied with the Power to Quit program, being African American was a significant

**Table 2. Participants' feedback on the patch, appointments, and counseling sessions.**

| Variable | Frequency | Percentage |
|---|---|---|
| **Patch** | | |
| **Number of weeks of self-reported adherence to patch Median (IQR) (n = 310)** | 6 (6–8) | |
| **Proportion of patch used (n = 314)** | | |
| None | 2 | 0.6 |
| Less than one half | 19 | 6.1 |
| About one half | 41 | 13.1 |
| More than one half | 39 | 12.4 |
| All or nearly all | 212 | 67.5 |
| Don't know/ Don't remember | 1 | 0.3 |
| **Experienced difficulties with using the patch** | | |
| **Yes** | 124 | 39.4 |
| **No** | 191 | 60.6 |
| **Primary difficulties encountered with the use of patch (n = 124)** | | |
| Remembering to put it on every day | 33 | 26.6 |
| Side effect: Burning and/or itching | 26 | 21.0 |
| Side effect: Difficulty sleeping | 20 | 16.1 |
| Did not reduce urge to smoke | 14 | 11.3 |
| Side effect: Nausea and/or vomiting | 12 | 9.7 |
| Carrying around/Not having a place to store them | 11 | 8.9 |
| Finding a clean spot on my skin | 8 | 6.4 |
| **Continued patch after study ended (n = 313)** | | |
| **Yes** | 144 | 45.7 |
| **No** | 169 | 53.99 |
| **Would have continued if given (n = 313)** | | |
| **Yes** | 257 | 81.6 |
| **No** | 169 | 53.99 |
| **Rated helpfulness of the patch (n = 313)** | | |
| Very unhelpful | 26 | 8.3 |
| Unhelpful | 9 | 2.9 |
| Not sure how helpful | 32 | 10.2 |
| Helpful | 73 | 23.3 |
| Very helpful | 173 | 55.3 |
| **Appointments** | | |
| **Primary reason for missing appointments (n = 189)** | | |
| Forgot | 75 | 39.7 |
| Work | 31 | 16.4 |
| Another appointment | 22 | 11.6 |
| No transportation | 18 | 9.5 |
| Out of town | 14 | 7.4 |
| Sick | 10 | 5.3 |
| In the hospital | 6 | 3.2 |
| Looking for housing | 5 | 2.7 |
| School or other classes | 5 | 2.7 |
| Job hunting or interview | 1 | 0.5 |
| Had not quit smoking | 1 | 0.5 |
| Don't feel like it, not in the mood to go | 1 | 0.5 |
| **What helped the most as appointment Reminders (n = 315)** | | |

(*Continued*)

**Table 2.** (Continued)

| Variable | Frequency | Percentage |
|---|---|---|
| Reminder phone calls or messages | 153 | 48.6 |
| Paper reminder clips | 59 | 18.7 |
| I didn't need help with this | 20 | 6.3 |
| Face to face reminders | 13 | 4.1 |
| All of them | 38 | 12.1 |
| None of them | 5 | 1.6 |
| Other | 27 | 8.6 |
| **Primary motivations for keeping appointments (n = 314)** | | |
| Wanted to quit smoking | 137 | 43.6 |
| Incentives (other than bus passes) | 98 | 31.2 |
| Liked the staff | 27 | 8.60 |
| Responsibility to see the counsellor | 10 | 3.18 |
| Community mobilizer encouragement | 8 | 2.55 |
| Enjoyed meetings/ sessions | 8 | 2.55 |
| Bus passes | 5 | 1.59 |
| Nothing else to do | 3 | 0.96 |
| Learned new information | 1 | 0.32 |
| Other | 17 | 5.41 |
| **Counseling** | | |
| **Perceived helpfulness of counseling sessions (n = 313)** | | |
| Very unhelpful | 22 | 7.0 |
| Unhelpful | 4 | 1.3 |
| Not sure how helpful | 27 | 8.7 |
| Helpful | 74 | 23.6 |
| Very helpful | 186 | 59.4 |
| **Would like to continue with counseling (n = 313)** | | |
| Yes | 200 | 63.9 |
| No | 100 | 32.0 |
| I don't know | 13 | 4.1 |

predictor of overall program satisfaction. The high satisfaction among African American participants in this program may be as a result of the involvement of racially concordant research staff in the design and implementation of the intervention. One of the two counselors that provided motivational interviewing was African American, two African Americans who had experienced homelessness in the past were recruited as participant mobilizers. Additionally, the members of the Community Advisory Board (CAB) included members who were familiar with the needs and desires of homeless people. Saha et al. found that provider-patient racial concordance can influence satisfaction with health care among African American and Hispanic populations [35]. Similarly, racial concordance in addition to shared experiences of homelessness may have positively influenced the study implementation and by extension, program satisfaction among the African American participants. We observed that age at smoking initiation may also be associated with overall program satisfaction. Younger age at smoking initiation has been associated with an increased likelihood of relapse [36]. This implies that adult smokers who started smoking at an earlier age might have more quit attempts and therefore may be more responsive to smoking cessation programs.

**Table 3. Program satisfaction among respondents.**

| Variable | Frequency | Percentage |
|---|---|---|
| **Overall program satisfaction** | | |
| Not satisfied at all | 2 | 0.7 |
| Somewhat unsatisfied | 1 | 0.3 |
| Neutral–don't feel strongly either way | 13 | 4.1 |
| Somewhat satisfied | 51 | 16.2 |
| Very satisfied | 248 | 78.7 |
| **Satisfaction with program location** | | |
| Not satisfied at all | 7 | 2.2 |
| Somewhat unsatisfied | 5 | 1.6 |
| Neutral–don't feel strongly either way | 14 | 4.4 |
| Somewhat satisfied | 33 | 10.5 |
| Very satisfied | 256 | 81.3 |
| **Satisfaction with program counselors (n = 314)** | | |
| Not satisfied at all | 5 | 1.6 |
| Somewhat unsatisfied | 2 | 0.6 |
| Neutral–don't feel strongly either way | 11 | 3.5 |
| Somewhat satisfied | 24 | 7.6 |
| Very satisfied | 272 | 86.4 |
| **Satisfaction with the schedule, i.e., appointment times, frequency and length of sessions** | | |
| Not satisfied at all | 2 | 0.6 |
| Somewhat unsatisfied | 7 | 2.2 |
| Neutral–don't feel strongly either way | 9 | 2.9 |
| Somewhat satisfied | 45 | 14.3 |
| Very satisfied | 252 | 80.0 |
| **Satisfaction with staff** | | |
| Not satisfied at all | 2 | 0.6 |
| Somewhat unsatisfied | 1 | 0.3 |
| Neutral–don't feel strongly either way | 10 | 3.2 |
| Somewhat satisfied | 33 | 10.5 |
| Very satisfied | 269 | 85.4 |
| **Feelings of trust in staff** | | |
| 1-Least | 2 | 0.6 |
| 2 | 2 | 0.6 |
| 3 | 7 | 2.2 |
| 4 | 30 | 9.5 |
| 5–Most | 274 | 87.0 |
| Mean (SD) | 4.8 (0.6) | |
| **Felt comfortable asking questions** | | |
| 1-Least | 0 | 0.0 |
| 2 | 1 | 0.3 |
| 3 | 5 | 1.6 |
| 4 | 25 | 7.9 |
| 5–Most | 284 | 90.2 |
| Mean (SD) | 4.9 (0.4) | |
| **Felt treated with respect** | | |
| 1-Least | 1 | 0.3 |

*(Continued)*

**Table 3.** (Continued)

| Variable | Frequency | Percentage |
|---|---|---|
| 2 | 2 | 0.6 |
| 3 | 3 | 0.9 |
| 4 | 14 | 4.4 |
| 5–Most | 295 | 93.7 |
| Mean (SD) | 4.9 (0.4) | |
| **Felt research was adequately explained** | | |
| 1-Least | 3 | 1.0 |
| 2 | 1 | 0.3 |
| 3 | 2 | 0.6 |
| 4 | 19 | 6.0 |
| 5–Most | 290 | 92.1 |
| Mean (SD) | 4.9 (0.5) | |
| **Felt information provided was kept confidential** | | |
| 1- Least | 2 | 0.6 |
| 2 | 0.0 | 0.0 |
| 3 | 13 | 4.1 |
| 4 | 18 | 5.7 |
| 5–Most | 282 | 89.5 |
| Mean (SD) | 4.8 (0.6) | |
| **Felt attention was given to special needs** | | |
| 1-Least | 1 | 0.3 |
| 2 | 4 | 1.3 |
| 3 | 7 | 2.2 |
| 4 | 24 | 7.7 |
| 5–Most | 277 | 88.5 |
| Mean (SD) | 4.8 (0.6) | |

Among smokers experiencing homelessness, financial incentives may increase smoking abstinence and quit attempts [37]. In our study, Visa cards, bus tokens and Subway restaurant coupons were the most preferred program incentive; likewise, the patch was rated as the most preferred program component. These findings are not surprising given that individuals experiencing homelessness often have limited access to food, transportation, employment, and health insurance [38]. We observed that participants' rankings of the various program components and incentives were not significantly associated with overall participant satisfaction. There may be several reasons as to why a significant association was not observed. First, approximately 25% of those initially enrolled in the study did not complete the week-26 feedback survey therefore subject attrition may have impacted the power to detect differences. Conversely, in previous studies, participants experiencing homelessness indicated high motivation and readiness to quit smoking [39, 40]. Therefore, incentives or specific components of the program may not have significantly affected participant motivation to participate in and complete the study, and subsequently their overall satisfaction with the program. Although most of the participants rated the patch as being 'very helpful,' many of the participants admitted that they often forgot to use the patch; phone calls and message reminders by the program staff were reported to be helpful. An understanding that smokers experiencing homelessness tend to have several competing priorities and building in novel ways to mitigate the effects of

**Table 4. The relationships between program satisfaction and other variables.**

| | Very Satisfied | Not very satisfied | Statistic | p-value |
|---|---|---|---|---|
| *Demographic characteristics* | | | | |
| **Age (mean ±SD) (n = 314)** | 45.8±9.7 | 45.4±10.3 | 0.26$^T$ | 0.794 |
| **Gender (n = 315)** | | | | |
| Male | 185(78.1) | 52(21.9) | 0.26 | 0.612 |
| Female | 63(80.8) | 15(19.2) | | |
| **Race/Ethnicity (n = 315)** | | | | |
| White | 76(71.7) | 30(28.3) | 5.80* | **0.055** |
| African American/Black | 155(83.3) | 31(16.7) | | |
| Other# | 17(73.9) | 6(26.1) | | |
| **Marital Status (n = 313)** | | | | |
| Married/Living with significant other | 12(75.0) | 4(25.0) | 0.21* | 0.900 |
| Divorced/Widowed/Separated | 106(78.5) | 29(21.5) | | |
| Never been married | 129(79.6) | 33(20.4) | | |
| **Highest level of Education (n = 315)** | | | | |
| < high school | 61(82.4) | 13(17.6) | 0.79* | 0.374 |
| ≥ high school | 187(77.6) | 54(22.4) | | |
| **Employment (n = 315)** | | | | |
| Currently employed | 26(83.9) | 5(16.1) | 0.54* | 0.461 |
| Currently unemployed | 222(78.2) | 62(21.8) | | |
| **Monthly income (n = 315)** | | | | |
| <$400 | 164(76.6) | 50(23.4) | 1.75* | 0.186 |
| ≥$400 | 84(83.2) | 17(16.8) | | |
| *Homelessness characteristic* | | | | |
| **Number of times homeless in past 3 years (n = 313)** | | | | |
| Once | 100(80.7) | 24(19.3) | 0.41* | 0.815 |
| One to three times(twice) | 64(77.1) | 19(22.9) | | |
| More than three times (≥3) | 83(78.3) | 23(21.7) | | |
| *Health and psychosocial characteristics* | | | | |
| **Self-reported general Health (n = 313)** | | | | |
| Excellent | 38(79.2) | 10(20.8) | 0.81* | 0.938 |
| Very Good | 68(76.4) | 21(23.6) | | |
| Good | 80(79.2) | 21(20.8) | | |
| Fair | 49(80.3) | 12(19.7) | | |
| Poor | 12(85.7) | 2(14.3) | | |
| **Depression PHQ9 ≥10 (mean±SD) (n = 313)** | 9.1±6.7 | 8.4±6.0 | 0.86* | 0.391 |
| **Stress PSS4, past 30 days, (mean±SD)** | 2.0±0.5 | 2.0±0.5 | -0.15$^T$ | 0.603 |
| *Smoking-related characteristics* | | | | |
| **Age started smoking regularly (mean ±SD) (n = 314)** | 16.2±5.7 | 17.8±7.5 | -1.80$^T$ | **0.073** |
| **Time to first cigarette** | | | | |
| ≤5 minutes | 117(80.1) | 29(19.9) | 0.32* | 0.571 |
| >5 minutes | 131(77.5) | 38(22.5) | | |
| **Number of 24 hour quit attempts past year, (mean±SD) (n = 310)** | 2.7±5.3 | 1.8±3.0 | 1.27$^T$ | 0.206 |
| **Confidence to quit, (mean±SD) (n = 315)** | 7.4±2.5 | 6.8±2.2 | 1.88$^T$ | **0.060** |
| *Substance abuse variables* | | | | |
| **Had a positive screen for alcohol dependence (n = 313)** | | | | |
| No | 104(81.9) | 23(18.1) | 1.38* | 0.240 |
| Yes | 142(76.3) | 44(23.7) | | |

*(Continued)*

**Table 4.** (Continued)

| | Very Satisfied | Not very satisfied | Statistic | p-value |
|---|---|---|---|---|
| **Had a positive screen for drug dependence (n = 314)** | | | | |
| No | 35(70.0) | 15(30.0) | 2.66* | 0.103 |
| Yes | 212(80.3) | 52(19.7) | | |
| **Top Ranked component(n = 315)** | | | | |
| Reading materials | 25(75.8) | 8(24.2) | 1.50* | 0.681 |
| Counselling sessions | 73(78.5) | 20(21.5) | | |
| The patch | 117(81.3) | 27(18.7) | | |
| Community mobilization | 33(73.3) | 12(26.7) | | |
| **Top Ranked incentive(n = 315)** | | | | |
| Visa gift cards | 155(79.90) | 39(20.10) | 1.57* | 0.456 |
| Bus tokens | 47(81.03) | 119(18.97) | | |
| Others* | 46(73.02) | 17(26.98) | | |

*Chi-square

T independent T-test

Overall satisfaction was dichotomized. not satisfied at all, somewhat unsatisfied, neutral and somewhat satisfied were coded as 0 (not very satisfied). Very satisfied was coded as 1(very satisfied)

#Other categories include America Indian/Alaskan Native, Hispanic, Filipino, Jewish, Indian, Irish, Multi-racial

competing priorities into the design of smoking cessation programs for smokers experiencing homelessness should be considered in future iterations.

## Limitations

This is one of the first studies to assess program satisfaction and participants preferences for the components of a smoking cessation program among smokers experiencing homelessness; however, it has some limitations. First, the sample was a convenience sample of individuals experiencing homelessness in the Twin Cities and may not be generalizable to persons

**Table 5. Predictors of program satisfaction among the respondents.**

| Variable | AOR(95% CI Lower limit, Upper limit) | p-value |
|---|---|---|
| Age (years) | 1.005(0.977, 1.035) | 0.695 |
| **Gender** | | |
| Female(ref) | | |
| Male | 0.832(1.035, 1.611) | 0.585 |
| **Race** | | |
| White (ref) | | |
| African American/Black | 1.847(1.027, 3.320) | **0.040*** |
| Others# | 1.018(0.356, 2.908) | 0.974 |
| **Age at smoking initiation** | 0.957(0.919, 0.998) | **0.040*** |
| **Confidence to quit smoking** | 1.108(0.989, 1.243) | 0.075 |

Pseudo R2 = 0.0370

Hosmer Lemeshow goodness of fit X^2 = 10.70 p = 0.220

#Other categories include America Indian/Alaskan Native, Hispanic, Filipino, Jewish, Indian, Irish, Multi-racial

*Significant at p<0.05

experiencing homelessness living in other cities. Future studies may wish to assess program satisfaction in other parts of the United States. Second, the survey responses may reflect some level of social desirability bias. Individuals, especially those who are low-income and those with low educational attainment, as may be the case with persons experiencing homelessness, tend to respond favorably regardless of content [41, 42]. Our research team constituted persons, who had experiences with homelessness either directly or indirectly; having individuals who could identify with the participants' background may have attenuated but not eliminated participants' tendency to give favorable responses. Third, non -response bias which arises when respondents are systematically different from non-respondents was addressed by instituting measures to minimize attrition. Daily reminder calls were made to participants during the week prior to appointments, until the window for completing appointments was closed. Reminder slips were also given to the respondents at the time of setting the appointment. Fourth, we did not explore the possible influence racial concordance may have played in participant's satisfaction. This should be considered in future research. Fifth, results are based on cross-sectional data and therefore we make no claims on causality based on the statistical design of the study. Sixth, our study did not examine the association between program satisfaction and program outcomes, primarily because the number of successful quitters in this study was too small to yield any meaningful statistically significant differences. Previously published results of the effectiveness of this intervention, adding motivational interviewing counseling to nicotine patch for smoking cessation among this population found no significant differences in verified seven-day abstinence rate at the end of follow up between the intervention group and the control group [43]. Future studies should assess this association to determine if program satisfaction impacts both short-term outcomes (i.e. retention) and long-term outcomes (i.e. cessation). Seventh, the instrument used to assess satisfaction has not been tested for reliability or validity in this population or any other population. Future studies should be aimed at creating a valid and reliable instrument to assess satisfaction in populations experiencing homelessness as their perceptions of care may be shaped by their experiences of homelessness and may be uniquely different from the general population [44]. Finally, 45.7% of participants continued to use the nicotine patch after the study concluded even though 81% of participants reported they would have continued to use the nicotine patch if they were given more. At the conclusion of the study, unused nicotine patches were donated to the medical clinic located within the largest shelter that served as a study site. Direct access to NRT has huge implications for long term smoking cessation efforts among populations experiencing homelessness. Unfortunately, logistical and procedural efforts to ensure NRT adherence once the study ended proved to be challenging. The mobile nature of this population made it difficult to monitor participant access to NRT and other quit aids once the study concluded. Future studies should consider ways to connect populations who experience homelessness to smoking cessation programs not only in the immediate area but the wider metropolitan area.

## Conclusions

Visa cards and bus tokens seem to be the preferred incentives for participation among this group of smokers. Participants preferred the nicotine patch to the counseling sessions; however, reminders for consistent patch use were needed Preferences for program incentives were unrelated to overall program satisfaction. Satisfaction has implications for retention and long-term outcomes therefore future smoking cessation programs for people experiencing homelessness should be designed to enhance satisfaction.

## Acknowledgments

The authors thank Jennifer Warren, PhD, and project staff Sharae Walker, Bonnie Houg, R'Gina Sellers, Casey Tuck, Abimbola Olayinka, Carolyn Warner, Carolyn Bramante, MD, Julia Davis, Pravesh Napaul, and Brandi White for their assistance with implementation of the project. The authors further acknowledge the directors of participating shelters, Dorothy Day Center, Our Savior's Shelter, Listening House, Union Gospel Mission, Naomi Family Center, and People Serving People and, finally, express gratitude to the members of the CAB and the study participants.

## Author Contributions

**Conceptualization:** Oluwakemi Ololade Odukoya, Ekland A. Abdiwahab.

**Formal analysis:** Oluwakemi Ololade Odukoya.

**Methodology:** Oluwakemi Ololade Odukoya, Ekland A. Abdiwahab.

**Writing – original draft:** Oluwakemi Ololade Odukoya, Ekland A. Abdiwahab, Tope Olubodun, Sunday Azagba, Folasade Tolulope Ogunsola, Kolawole S. Okuyemi.

**Writing – review & editing:** Oluwakemi Ololade Odukoya, Ekland A. Abdiwahab, Tope Olubodun, Sunday Azagba, Folasade Tolulope Ogunsola, Kolawole S. Okuyemi.

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
