## [Decision Letter · Decision Letter 0]

14 Oct 2021

PONE-D-21-22291Implementing a smoking cessation intervention for homeless smokers: Participants preferences, feedback, and satisfaction   with the ‘power to quit’ programmePLOS ONE

Dear Dr. Odukoya,

Thank you for submitting your manuscript to PLOS ONE. After careful consideration, we feel that it has merit but does not fully meet PLOS ONE’s publication criteria as it currently stands. Therefore, we invite you to submit a revised version of the manuscript that addresses the points raised during the review process.

The manuscript has been evaluated by two reviewers, and their comments are available below.

The reviewers have raised a number of concerns that need attention. They request additional information on methodological aspects of the study, including details of the study setting and participant recruitment, as well as additional information about the development and execution of the survey.

Could you please revise the manuscript to carefully address the concerns raised?

We look forward to receiving your revised manuscript.

Kind regards,

Marianne Clemence

Associate Editor

PLOS ONE

Journal Requirements:

2. Please ensure you have included the registration number for the clinical trial referenced in the manuscript.

Reviewers' comments:

Reviewer's Responses to Questions

**Comments to the Author**

1. Is the manuscript technically sound, and do the data support the conclusions?

Reviewer #1: No

Reviewer #2: Yes

2. Has the statistical analysis been performed appropriately and rigorously? 

Reviewer #1: Yes

Reviewer #2: Yes

3. Have the authors made all data underlying the findings in their manuscript fully available?

Reviewer #1: Yes

Reviewer #2: Yes

4. Is the manuscript presented in an intelligible fashion and written in standard English?

Reviewer #1: Yes

Reviewer #2: Yes

5. Review Comments to the Author

Reviewer #1: The paper presents data on homeless participants’ satisfaction and preferences for the Power to Quit study from a survey conducted at 26 weeks. The questions asked in the survey are similar to a process evaluation of a trial which can be useful to help inform future studies and interventions. The resulting data will be useful feedback to those involved in the PTQ study, however, I have some concerns about the methods and whether the findings provide enough insight to make a significant contribution to the literature or for others designing future interventions with homeless populations. Additionally,

The data appears to be based on a 26-week feedback survey. Currently there is little description of the survey in the methods section. How were participants invited to take part, what was the method of administration/data collection, was there a participant incentive for completing the survey, how long was the survey? The methods would also benefit from a full description of what the survey covered. The results suggest that more was asked than is currently outlined in the methods.

Was all the data used in the paper collected at the 26-week survey? It would be useful to clarify this, i.e. was the demographic and other data collected at 26-weeks or was this collected earlier in the trial? This is important to highlight because some of the smoking data, if collected at 26 weeks, will likely have been influenced by participation in the trial, for example the % reporting an unsuccessful quit attempt in the last year and the high confidence to quit score could have been a direct result of the trial, and not therefore suggestive of homeless populations, and should be noted.

How was the survey (and importantly, the questions) developed? Were they tested for participants’ understanding and relevance? There are potential issues with some of the main questions asked. For example, the question, ‘Which items were most helpful?’ This question is open to interpretation. Did the authors mean which items were most helpful for reducing smoking or quitting? The inclusion of ‘reading materials’ or ‘community mobile contacts’ suggest that the question could also have been interpreted as ‘most helpful sources of information?’

The monetary/voucher incentives were particularly valued by homeless study participants. This is unsurprising, although they were not linked with programme satisfaction. The authors write that “incentives or specific components of the programme may not have significantly affected participant motivation to participate in and complete the study”. What were the incentives in the study used for, e.g. enrolment, attending follow-up appointments, motivation for smoking abstinence? This needs to be clarified as currently the paper doesn’t tell us very much about how incentives were used in the trial, and therefore the implications for future interventions are limited.

Reviewer #2: Thank you for the opportunity to review this paper. The manuscript describes satisfaction with the Power to Quit smoking cessation intervention in Minneapolis and St. Paul, Minnesota. I thought the manuscript was interesting and had practical recommendation for people conducting cessation trials with people experiencing homelessness. I did think that findings could probably be summarized as a brief report if that was an option. I have several other suggestions below:

1) I might consider changing the language to reflect people or populations experiencing homelessness or people experiencing homelessness who smoke as an indicator that this population is constantly changing and is dynamic based on economic and other circumstances, and move away from calling the population, "the homeless" or "homeless smokers" which signifies a static condition and potentially a character trait (which we know it is not). This might be viewed as less stigmatizing. I would suggest making this change throughout the manuscript.

2) In the abstract, I might include how satisfaction was qualitatively assessed. I might bring in the point that the rate of satisfaction was high among African American participants in part because there may have been race concordance between study staff, community advisory board and AA participants. I think that is the salient point here that it is helpful to have staff that have lived experiences of participant in the study to improve the effectiveness of the study.

3) Methods: I might include where the study sites were in Minneapolis and St. Paul - I recall they were in shelters in these cities. Did participants have to commute? What about timings for the assessment and counseling sessions -- were they flexible?

4) The authors alluded to this but was the validity of the satisfaction question in this sample -- could the authors conduct an internal validity of this question in this sample?

5) Page 7, Line 150 -- which item was most helpful to you -- is this in relation to smoking cessation or general life?

6) The authors described many covariates, e.g., health and psychosocial covariates, smoking-related covariates, substance use and dependence covariates -- while these are interesting, I wonder how much is relevant to the analysis on satisfaction as none of these variables are controlled for in the model. I might consider taking these out of this paper and referring to the original Power to Quit study findings for descriptive statistics. It does not seem that satisfaction differed based on mental health or substance use based on these findings and if so, might clarify and explain why that might be in the results and discussion.

7) A few questions on the results/discussion:

- I might consider discussing in the results, how satisfaction varied for participants who were abstinent at 6 months vs. not; for participants in the intervention vs control (I did not see this distinction and it seems relevant)

- one of the participants' feedback was on continuing patch and counseling if that option was available to participants -- and it has implications for extended duration interventions for PEH, perhaps it might be good to discuss this point in the discussion.

- when I read that African American/Black participants had higher satisfaction levels, I immediately thought this was because staff might have been African American/Black. Wonder if there was a way to create a variable on racially concordant staff/participant dyad/group variable vs. not and see if that is associated with satisfaction. I think this finding is really important for anyone doing work among PEH, as African American/Black folks are over-represented in populations experiencing homelessness in large urban cities, and calls for having study staff/teams that are representative of the lived experiences and gender/racial.ethnic diversity of PEH.

- can the authors comment on when should satisfaction be ascertained (e.g., at what time points -- all study visits), and how can intervention design be malleable to changing views on satisfaction during an intervention among PEH.

6. PLOS authors have the option to publish the peer review history of their article (what does this mean?). If published, this will include your full peer review and any attached files.

Reviewer #1: No

Reviewer #2: No

---

## [Author Response · Author response to Decision Letter 0]

17 Dec 2021

Professor Joerg Heber 

Editor-in-Chief 

PLOS ONE

26th November 2021

PONE-D-21-22291

Implementing a smoking cessation intervention for homeless smokers: Participants preferences, feedback, and satisfaction with the ‘power to quit’ programme

We would like to thank the academic editor and the reviewers for their constructive feedback and helpful comments on our manuscript titled ‘Implementing a smoking cessation intervention for homeless smokers: Participants preferences, feedback, and satisfaction with the ‘power to quit’ programme’. Please see below our responses. 

JOURNAL REQUIREMENTS:

Comment 1. Please ensure that your manuscript meets PLOS ONE's style requirements, including those for file naming. 

Our Response (1): This manuscript meets PLOS ONE's style requirements

Comment 2. Please ensure you have included the registration number for the clinical trial referenced in the manuscript.

Our Response (2): Trial Registration number has been stated (Line 111)

Comment 3. We note that you have stated that you will provide repository information for your data at acceptance. Should your manuscript be accepted for publication, we will hold it until you provide the relevant accession numbers or DOIs necessary to access your data. If you wish to make changes to your Data Availability statement, please describe these changes in your cover letter and we will update your Data Availability statement to reflect the information you provide.

Our Response (3): Data will be made available on request. This is now being included in the cover letter.

Comment 4. Please include your full ethics statement in the ‘Methods’ section of your manuscript file. In your statement, please include the full name of the IRB or ethics committee who approved or waived your study, as well as whether or not you obtained informed written or verbal consent. If consent was waived for your study, please include this information in your statement as well. 

Our Response (4): Full name of the IRB as well as statement on informed consent has been stated. Study procedures were approved and monitored by the Institutional Review Board of the University of Minnesota Medical School (Study Number: 1307M39761). Written informed consent was obtained from participants prior to data collection. (Lines 306 - 308)

REVIEWER 1

Comment 1a: Currently, there is little description of the survey in the methods section. How were participants invited to take part.

Our Response (1a): ‘Researchers recruited participants by conducting health fairs, holding informational interviews, posting flyers, and announcing the study at homeless shelters. Participants also helped to recruit by word of mouth.’ This has been stated in the methods section (Line 148-150) The eligibility criteria is also stated (Lines 150 – 167)

Comment 1b: What was the method of administration/data collection, 

Our Response (1b): Validated questionnaires were used to collect survey data. Metric measurements of height and weight were collected to calculate body mass index. Carbon monoxide (CO) and saliva cotinine was assessed as biomarkers of tobacco use. The participants were asked to exhale into a carbon monoxide monitor. Patch adherence was measured by: ‘patch checks’ (visual verification of whether a participant was wearing a patch); ‘patch counts’ (documenting the number of patches left in the participant’s possession); and administration of the Morisky scale, a self-reported adherence scale modified to assess adherence to NRT patch 

Detailed method of data collection has been stated in the methods (Lines 170 – 196)

Comment 1c: was there a participant incentive for completing the survey, how long was the survey? 

Our Response (1c): Incentives were given as compensation for participants time and effort. At each of the 15 visits, participants received incentives. For attending the final 6-month visit, participants received a $40 gift card and a sweatshirt. For participants who attended all 15 sessions, the monetary incentives totaled $275 over 6 months. The study lasted for 26 months. (Details in Lines 131-140)

Comment 1d: The methods would also benefit from a full description of what the survey covered. The results suggest that more was asked than is currently outlined in the methods.

Our Response (1d): The methods section has been improved upon and contains information on:

An overview of the study, aim of the study and the intervention components (Line 111-140)

Recruiting of participants and eligibility criteria ( Lines 148 -167)

Data collection (Lines 170 -196)

Comment 2: Was all the data used in the paper collected at the 26-week survey? It would be useful to clarify this, i.e. was the demographic and other data collected at 26-weeks or was this collected earlier in the trial? This is important to highlight because some of the smoking data, if collected at 26 weeks, will likely have been influenced by participation in the trial, for example the % reporting an unsuccessful quit attempt in the last year and the high confidence to quit score could have been a direct result of the trial, and not therefore suggestive of homeless populations, and should be noted.

Our Response (2): Study data was collected at baseline, week 1, week 2, week 4, week 6, week 8, week 10, week 12, week 14, week 16, week 18, week 20, week 24, and week 26. However, data analyzed in this study was limited to the baseline and week 26 feedback survey. The detailed information is provided in the Data collection section (Lines 179-196)

Comment 3: How was the survey (and importantly, the questions) developed? Were they tested for participants’ understanding and relevance? There are potential issues with some of the main questions asked. For example, the question, ‘Which items were most helpful?’ This question is open to interpretation. Did the authors mean which items were most helpful for reducing smoking or quitting? The inclusion of ‘reading materials’ or ‘community mobile contacts’ suggest that the question could also have been interpreted as ‘most helpful sources of information?’

Our Response (3):The survey tool was developed by the research team with the aim of assessing participants satisfaction with the research program and its components. Prior to the start of the survey, all questions were pretested for understanding and relevance. The survey was administered by trained interviewers to avoid possible misinterpretations of the questions.

Comment 4: The monetary/voucher incentives were particularly valued by homeless study participants. This is unsurprising, although they were not linked with programme satisfaction. The authors write that “incentives or specific components of the programme may not have significantly affected participant motivation to participate in and complete the study”. What were the incentives in the study used for, e.g. enrolment, attending follow-up appointments, motivation for smoking abstinence? This needs to be clarified as currently the paper doesn’t tell us very much about how incentives were used in the trial, and therefore the implications for future interventions are limited.

Our Response (4): Incentives were given as compensation for participants time and effort. Incentives were provided at each of the 15 visits. They were provided when participants completed surveys, attended brief retention visits, attended treatment visits, and at the follow-up 6 month visit. (Details in Lines 131-140) 

REVIEWER 2

Comment 1: I might consider changing the language to reflect people or populations experiencing homelessness or people experiencing homelessness who smoke as an indicator that this population is constantly changing and is dynamic based on economic and other circumstances, and move away from calling the population, "the homeless" or "homeless smokers" which signifies a static condition and potentially a character trait (which we know it is not). This might be viewed as less stigmatizing. I would suggest making this change throughout the manuscript.

Our Response (1): The language has been changed to reflect populations experiencing homelessness or people experiencing homelessness.

Comment 2: In the abstract, I might include how satisfaction was qualitatively assessed. I might bring in the point that the rate of satisfaction was high among African American participants in part because there may have been race concordance between study staff, community advisory board and AA participants. I think that is the salient point here that it is helpful to have staff that have lived experiences of participant in the study to improve the effectiveness of the study.

Our Response(2): The assessment of satisfaction has been included in the abstract (Line 32).

 Race concordance influencing satisfaction has been included in the abstract (Lines 49 – 52) 

Race concordance influencing satisfaction has been included in the discussion ab-initio (Line 415 – 429) 

Comment 3 Methods: I might include where the study sites were in Minneapolis and St. Paul - I recall they were in shelters in these cities. Did participants have to commute? What about timings for the assessment and counseling sessions -- were they flexible?

Our Response(3): Study sites were situated within homeless shelters and/or homeless service centers that are located in the downtown/city center easily accessible by public transportation such as city-operated buses and light rail. (Line 146 – 148) 

Because of the nature of the study population, timings for assessments and counselling sessions were flexible. (Line 180 – 181)

Comment 4: The authors alluded to this but was the validity of the satisfaction question in this sample -- could the authors conduct an internal validity of this question in this sample?

Our Response(4): The importance of satisfaction is widely recognized in substance use treatment therefore future interventions should aim to assess the internal validity of this specific construct.(Line 484-489) Unfortunately, the Power to Quit study concluded in 2011 therefore we are unable to assess the internal validity of this question within our study sample. In addition, internal validation of the satisfaction question was not planned a priori therefore the satisfaction questions were not constructed in a way that would allow us to go back and run statistical analysis such as confirmatory factor analysis. 

Comment 5: Page 7, Line 150 -- which item was most helpful to you -- is this in relation to smoking cessation or general life?

Our Response(5): We thank the reviewers for this comment. The question items that asked participants to state the items they found most helpful was with respect to smoking cessation. The survey was administered by trained interviewers who were able to provide additional context to questions where needed. The respondents were aware that the questions asked were with respect to smoking cessation. 

Comment 6: The authors described many covariates, e.g., health and psychosocial covariates, smoking-related covariates, substance use and dependence covariates -- while these are interesting, I wonder how much is relevant to the analysis on satisfaction as none of these variables are controlled for in the model. I might consider taking these out of this paper and referring to the original Power to Quit study findings for descriptive statistics. It does not seem that satisfaction differed based on mental health or substance use based on these findings and if so, might clarify and explain why that might be in the results and discussion.

Our Response(6): We included information on the covariates that were used in the analysis of this paper. We have expunged information other covariates (self-efficacy to refrain from smoking) that were not included in the descriptive and bivariate analysis. 

In multivariable analysis, variables significant at p<0.10 in the bivariate analysis were imputed in the logistic regression (Line 298-299 ) We feel it is important to show that some of the covariates did not show statistically significant association with satisfaction. Hence, we have included table 4 showing the bivariate analysis (Line 395).

Comment 7a: On the results/discussion, I might consider discussing in the results, how satisfaction varied for participants who were abstinent at 6 months vs. not; for participants in the intervention vs control (I did not see this distinction and it seems relevant)

Our Response(7a): The number of participants who successfully quit at the end of the survey was very small. As a result, such statistical analysis would not have yielded any meaningful statistically significant differences. We have highlighted this as a study limitation and highlighted it as an area for future research.(Lines 476-484)

Comment 7b: On the results/discussion, one of the participants' feedback was on continuing patch and counseling if that option was available to participants -- and it has implications for extended duration interventions for PEH, perhaps it might be good to discuss this point in the discussion.

Our Response(7b): Thank you for this recommendation. We have updated our results and discussion sections to include this discussion point. (Table 2, Line 489 – 500)

Comment 7c: On the results/discussion - when I read that African American/Black participants had higher satisfaction levels, I immediately thought this was because staff might have been African American/Black. Wonder if there was a way to create a variable on racially concordant staff/participant dyad/group variable vs. not and see if that is associated with satisfaction. I think this finding is really important for anyone doing work among PEH, as African American/Black folks are over-represented in populations experiencing homelessness in large urban cities, and calls for having study staff/teams that are representative of the lived experiences and gender/racial.ethnic diversity of PEH.

Our Response(7c): Thank you for this comment. While we agree that this would have enriched the study findings, we did not set out to assess this relationship in the initial study. As a result, we are unable to generate a variable for racially concordant staff/participant dyad/group and include this in our analysis. However, we have included this point as an area for future research.

 (Line 472 – 474 )

Comment 7d: Can the authors comment on when should satisfaction be ascertained (e.g., at what time points -- all study visits), and how can intervention design be malleable to changing views on satisfaction during an intervention among PEH.

Our Response(7d): Satisfaction should be at the least assessed at the end of the study. The pros and cons of assessing satisfaction at multiple time points should be balanced. The pro of assessing satisfaction is the ability to obtain more information that will help to inform future studies. This would be appropriate in a study with a pragmatic design where design changes are built apriori on the study protocol. However, repeated measures of satisfaction may have unintended consequences. Participants may be more prone to giving socially desirable responses, the repeated surveys may become too burdensome for participants and may negatively impact attrition, and the repeated measures may become too burdensome for the research team. Furthermore, changing views on satisfaction should be incorporated into the study without changing randomization of the participants and without creating barriers to carrying out the study. 

Yours sincerely, 

Oluwakemi Ololade Odukoya

On behalf of the study investigators

---

## [Decision Letter · Decision Letter 1]

15 Feb 2022

PONE-D-21-22291R1Implementing a smoking cessation intervention for people experiencing homelessness: participants’ preferences, feedback, and satisfaction with the ‘power to quit’ programPLOS ONE

Dear Dr. Odukoya,

Thank you for submitting your manuscript to PLOS ONE. After careful consideration, we feel that it has merit but does not fully meet PLOS ONE’s publication criteria as it currently stands. Therefore, we invite you to submit a revised version of the manuscript that addresses the points raised during the review process.

Thank you for submitting this revised manuscript to PLOS ONE and addressing the previous reviewers’ comments. The revised manuscript is well written and addresses an important topic for public health and clinical research. A third internal statistical reviewer was solicited for comments as you can see below. Please address this reviewer’s new comments or provide a rationale to not make the recommended changes (except in the instances where I do not feel the new recommendations are necessary to improve the manuscript, below). In addition, as I am a new editor for this manuscript and did not review the initial submission myself, I offer several comments that I hope will enhance the manuscript.

First, the Data Availability field states that Yes all data are fully available without restriction, but this does not appear to be the case per the cover letter. If data are only available on request, please state why there are legal or ethical restrictions on sharing data. The doi in dryad does not lead to public data set. Please review the PLOS ONE FAQ regarding data sharing: https://journals.plos.org/plosone/s/data-availability

Line 58: 20% is outdated if referring to U.S, which is now around 14%.

Line 292-293: Please rephrase to clarify that participants had a positive screen for alcohol or drug dependence rather than identifying them as alcohol/drug dependent (given the relationship between screeners and actual diagnostic status of substance use disorder is far from 100%). Same guidance for Table 1.

Table 1: Please relabel rows to 1, 2, >3. It seems odd and minimizing to use the word “just” as a qualifier for the number of times experiencing homelessness.

Table 3: Overall program satisfaction – please report frequency/percentage for all 5 response options.

Table 4: In table note please remind the reader of how ‘satisfied’ and ‘not satisfied’ were coded. Why was ‘somewhat satisfied’ coded as ‘not satisfied’? I can see a justification for combining response options 1-3, but ‘somewhat satisfied’ doesn’t seem like it fits under the umbrella of ‘not satisfied’. I won't require to re-code and rerun all analyses, but please provide a justification for this decision if you prefer to maintain this coding scheme, but adjust labels (e.g., ‘Very satisfied vs No very satisfied’. I recognize these satisfaction outcome data are highly skewed, but this should be justified in the methods and mentioned as a qualification in the Discussion.

Table 4: Please justify your alpha of 0.10 for Table 4.

Table 5: Could include the 95% CI in parentheses after the AOR (e.g., 1.005 (0.977, 1.035)). Please label as 95% CI in table note or column header (per Reviewer 3).

Reviewer 3 provides useful suggestions that I recommend you incorporate into the revised manuscript. Given that this manuscript has already been reviewed by two peer reviewers, I would not require you to make the following changes recommended by Reviewer 3:

-Participants (line 150): readers can refer to your citation of the main design/outcomes papers

-Data Collection (Line 176-178): you already state only baseline/26wk are presented here.

-Table 1: do not need to recode age. Cosmetic changes to tables will occur during copy-editing, not necessary at this stage.

We look forward to receiving your revised manuscript.

Kind regards,

Jesse T. Kaye, PhD

Academic Editor

PLOS ONE

Journal Requirements:

Reviewers' comments:

Reviewer's Responses to Questions

**Comments to the Author**

1. If the authors have adequately addressed your comments raised in a previous round of review and you feel that this manuscript is now acceptable for publication, you may indicate that here to bypass the “Comments to the Author” section, enter your conflict of interest statement in the “Confidential to Editor” section, and submit your "Accept" recommendation.

Reviewer #1: All comments have been addressed

Reviewer #2: All comments have been addressed

Reviewer #3: (No Response)

2. Is the manuscript technically sound, and do the data support the conclusions?

Reviewer #1: Yes

Reviewer #2: Yes

Reviewer #3: Partly

3. Has the statistical analysis been performed appropriately and rigorously? 

Reviewer #1: Yes

Reviewer #2: Yes

Reviewer #3: No

4. Have the authors made all data underlying the findings in their manuscript fully available?

Reviewer #1: Yes

Reviewer #2: Yes

Reviewer #3: Yes

5. Is the manuscript presented in an intelligible fashion and written in standard English?

Reviewer #1: Yes

Reviewer #2: Yes

Reviewer #3: Yes

6. Review Comments to the Author

Reviewer #1: (No Response)

Reviewer #2: (No Response)

Reviewer #3: The manuscript entitled ‘Implementing a smoking cessation intervention for people experiencing homelessness participants’ preferences, feedback, and satisfaction with the ‘power to quit’ program’ with the aim to assess participants' satisfaction and preferences for the Power to Quit (PTQ) program.

The manuscript could be further improved based on the following comments.

Materials and methods

Participants

Line 150, more information on the randomization method, blinding, allocation concealment to be provided.

Data Collection

Line 176-178, to state that these data in week 2,4,6,8 will not be presented in this manuscript.

Exposure variables

Line 195, coding labelling to be fully provided e.g. 1(most helpful), 2( ), 3(not helpful). Likewise Line 201 e.g. 1(liked the most), 2( ), 3(did not like it)

Line 264, Analysis to be written as Statistical analyses.

Line 269, for the statement ‘In addition, we controlled for study arm,’ the reason to analyze as combined groups and not groups comparison to be clearly stated before displaying the results. Would be good to include a description on missing data.

Results

Line 284, full name for GED to be provided.

Line 322. 21% to be replaced with 21.0%.

Line 324, Over 80% to replaced with 81.6%

Line 337, typo ‘thatthe’

Line 303 Table 1, title too brief. For monthly income, the symbol for the income category is incorrect (different to Line 214). For the homelessness characteristics ‘just once, One to three times(twice) and More than three times (≥3)’ are confusing Perhaps just state once, twice, thrice or more. Age started smoking regularly could have categorized as <10, 10-19, 20-29, 30-39 etc . Drug dependent and not drug dependent to be unbold.

Line 303 Table 1 and Line 330 Table 2, all the data to be presented (i.e No to be included) while missing data to be denoted in the table footnote and n to be stated for all variables.

The tables require cosmetic changes and the variables to be clearly separated with a space for easy identification. Italicized or unitalicized to be consistent for the variables.

Line 356 Table 3, the figure 4 or 5 in the subcategory to be spelled out or denoted in the table footnote. n to be stated and any missing data to be denoted.

Line 360 Table 4, n to be stated for each variable. Actual symbol chi-square X^2 to be used. The chi square value to be reduced to 2 decimal points and the decimal points for p value to be standardized. All the statistical tests used in Table 4 to be denoted in the table footnote. For the age, depression, stress, age started smoking regularly, number of 24 hour quit attempts past year, confidence to quit variables, the data were presented as mean± sd. The statistical test to be stated. If chi-square test was employed, the categories of each variable and frequency to be displayed. For the self-reported general health category poor, 14.2% to be replaced with 14.3%. Please re check the chi-square value and p value for the variable ‘top ranked incentive’.

The word p value or p-value to be consistent with Table 5. # was mentioned in table footnote but the label nowhere found in the table.

Line 373 Table 5, the model summary such as pseudo R^2 and goodness of fit test to be provided. 95%CI to be stated before lower and upper limit are stated.

Line 367, the sentence ‘Race was with a predictor of overall programme satisfaction.’ requires revision.

Line 376, p=.05 to be replaced with p < 0.05.

Discussion

Line 427, 435 & Line 437, typos thesurvey, Fourth,, Fifth..

Was there any other possible bias arising from the interviewing process?

7. PLOS authors have the option to publish the peer review history of their article (what does this mean?). If published, this will include your full peer review and any attached files.

Reviewer #1: No

Reviewer #2: No

Reviewer #3: No

---

## [Author Response · Author response to Decision Letter 1]

30 Mar 2022

Professor Joerg Heber 

Editor-in-Chief 

PLOS ONE

25th March 2022

PONE-D-21-22291

Implementing a smoking cessation intervention for homeless smokers: Participants preferences, feedback, and satisfaction with the ‘power to quit’ programme

We would like to thank the academic editor and reviewer for their constructive feedback and helpful comments on our manuscript titled ‘Implementing a smoking cessation intervention for homeless smokers: Participants preferences, feedback, and satisfaction with the ‘power to quit’ programme’. Please see below our responses. 

EDITORS COMMENTS:

Comment 1. First, the Data Availability field states that Yes all data are fully available without restriction, but this does not appear to be the case per the cover letter. If data are only available on request, please state why there are legal or ethical restrictions on sharing data. The doi in dryad does not lead to public data set. Please review the PLOS ONE FAQ regarding data sharing: https://journals.plos.org/plosone/s/data-availability

Our Response (1): All data are fully available without restriction an can be accessed at https://www.openicpsr.org/openicpsr/project/165321/version/V1/view This has also been stated in the cover letter

Comment 2. Line 58: 20% is outdated if referring to U.S, which is now around 14%.

Our Response (2): The information on the prevalence of smoking in the general population in the US has been updated (Line 59). The reference has also been updated (Line 520-522). 

Comment 3. Line 292-293: Please rephrase to clarify that participants had a positive screen for alcohol or drug dependence rather than identifying them as alcohol/drug dependent (given the relationship between screeners and actual diagnostic status of substance use disorder is far from 100%). Same guidance for Table 1.

Our Response (3): Thank you for your comment. The term “Alcohol/drug dependent” has been re-phrased as “had a positive screen for alcohol or drug dependence” (Line 297-298, Table 1 Line 307, Table 4 Line 369 ) 

Comment 4. Table 1: Please relabel rows to 1, 2, >3. It seems odd and minimizing to use the word “just” as a qualifier for the number of times experiencing homelessness.

Our Response (4): The rows have been re-labelled as Once, Twice, Thrice (Table 1 Line 307, Table 4 Line 369 ) 

Comment 5: Table 3: Overall program satisfaction – please report frequency/percentage for all 5 response options.

Our Response (5): The frequency/percentage values have been reported for all 5 responses. (Table 3 Line 362)

Comment 6: Table 4: In table note please remind the reader of how ‘satisfied’ and ‘not satisfied’ were coded. Why was ‘somewhat satisfied’ coded as ‘not satisfied’? I can see a justification for combining response options 1-3, but ‘somewhat satisfied’ doesn’t seem like it fits under the umbrella of ‘not satisfied’. I won't require to re-code and rerun all analyses, but please provide a justification for this decision if you prefer to maintain this coding scheme, but adjust labels (e.g., ‘Very satisfied vs No very satisfied’. 

Our Response (6): The options for satisfaction were 1)Not satisfied at all; 2) Somewhat unsatisfied; 3) Neutral; 4) Somewhat satisfied; 5) Very satisfied. There was no option stated as “somewhat satisfied” rather this was “somewhat unsatisfied” We believe that the options Not satisfied at all and somewhat unsatisfied both reflect some form of dissatisfaction, hence they were categorized together as “Not very satisfied” We have reminded the readers of how overall program satisfaction was re-coded. The labels have been adjusted as ‘Very satisfied vs Not very satisfied’ (Table 4 Line 369, Line 373-375)

Comment 7: Table 4: Please justify your alpha of 0.10 for Table 4.

Our Response (7): In multivariable analysis, variables significant at p<0.10 in the bivariate analysis were imputed in the logistic regression in order to include variables tending towards a positive significance. (Line 271-272)

Comment 8: Table 5: Could include the 95% CI in parentheses after the AOR (e.g., 1.005 (0.977, 1.035)). Please label as 95% CI in table note or column header (per Reviewer 3).

Our Response (8): The 95% CI is stated in parentheses after the AOR. The column header is labeled as AOR(95% CI Lower limit , Upper limit) (Table 5 Line 393)

Comment 9: Reviewer 3 provides useful suggestions that I recommend you incorporate into the revised manuscript. Given that this manuscript has already been reviewed by two peer reviewers, I would not require you to make the following changes recommended by Reviewer 3:

-Participants (line 150): readers can refer to your citation of the main design/outcomes papers

-Data Collection (Line 176-178): you already state only baseline/26wk are presented here.

-Table 1: do not need to recode age. Cosmetic changes to tables will occur during copy-editing, not necessary at this stage.

Our Response (9): Comments well taken

REVIEWER 3

Comment 1: Exposure variables

Line 195, coding labelling to be fully provided e.g. 1(most helpful), 2( ), 3(not helpful). Likewise Line 201 e.g. 1(liked the most), 2( ), 3(did not like it)

Our Response (1): Thank you for your comment. This has been done. (Line 197, Line 203-204)

Comment 2: Line 264, Analysis to be written as Statistical analyses.

Our Response (2): Thank you for your comment. This has been done. (Line 266)

Comment 3: Line 269, for the statement ‘In addition, we controlled for study arm,’ the reason to analyze as combined groups and not groups comparison to be clearly stated before displaying the results. 

Our Response (3):We thank the reviewer for this comment. Study arm was not controlled for in the regression model. This has been corrected in the methods (Statistical analysis) (Line 272)

Comment 4: Results

Line 284, full name for GED to be provided. 

Our Response (4): This has been done (Line 288-289).

Comment 5: Line 322. 21% to be replaced with 21.0%.

Our Response (5): This has been done (Line 327)

Comment 6: Line 324, Over 80% to replaced with 81.6% 

Our Response (6): This has been done (Line 329)

Comment 7: Line 337, typo ‘thatthe’

Our Response (7): This has been done (Line 343)

Comment 8: Line 303 Table 1, title too brief. For monthly income, the symbol for the income category is incorrect (different to Line 214). For the homelessness characteristics ‘just once, One to three times(twice) and More than three times (≥3)’ are confusing Perhaps just state once, twice, thrice or more. Drug dependent and not drug dependent to be unbold.

Our Response (8): This has been done (Table 1 Line 307 - 308)

Comment 9: Line 303 Table 1 and Line 330 Table 2, all the data to be presented (i.e No to be included) while missing data to be denoted in the table footnote and n to be stated for all variables.

Our Response (9): All the data has been presented. However, missing data was not shown in table, rather ‘n’ was stated for each variable. (Table 1 Line 308, Table 2 Line 336)

 Comment 10: Line 356 Table 3, the figure 4 or 5 in the subcategory to be spelled out or denoted in the table footnote. n to be stated and any missing data to be denoted.

Our Response (10): Thank you for this comment. This has been effected (Table 3 Line 362).

Comment 11: Line 360 Table 4, n to be stated for each variable. Actual symbol chi-square X^2 to be used. The chi square value to be reduced to 2 decimal points and the decimal points for p value to be standardized. All the statistical tests used in Table 4 to be denoted in the table footnote. For the age, depression, stress, age started smoking regularly, number of 24 hour quit attempts past year, confidence to quit variables, the data were presented as mean± sd. The statistical test to be stated. If chi-square test was employed, the categories of each variable and frequency to be displayed. For the self-reported general health category poor, 14.2% to be replaced with 14.3%. Please re check the chi-square value and p value for the variable ‘top ranked incentive’.

Our Response (11): The comments have been effected (Table 4 Line 370)

Comment 12: The word p value or p-value to be consistent with Table 5. # was mentioned in table footnote but the label nowhere found in the table.

Our Response (12): This has been done (Table 5 Line 393)

Comment 13: Line 373 Table 5, the model summary such as pseudo R^2 and goodness of fit test to be provided. 95%CI to be stated before lower and upper limit are stated.

Our Response (13): pseudo R^2 and goodness of fit test have been provided (Table 5 Line 393)

Comment 14: Line 367, the sentence ‘Race was with a predictor of overall programme satisfaction.’ requires revision.

Our Response (14): This has been corrected (Line 382)

Comment 15: Line 376, p=.05 to be replaced with p < 0.05.

Our Response (15): This has been corrected (Line 398)

Comment 16: Discussion

Line 427, 435 & Line 437, typos thesurvey, Fourth,, Fifth.. 

Our Response (16): This has been corrected (Line 447, 455, 457)

Comment 17: Was there any other possible bias arising from the interviewing process?

Our Response (17): There is also a possibility of non -response bias which arises when respondents are systematically different from non-respondents. Prior to data collection, measures were instituted to limit such biases and this has been included in the text. (Line 455-460).

Yours sincerely, 

Oluwakemi Ololade Odukoya

On behalf of the study investigators

---

## [Editor Report · Decision Letter 2]

5 May 2022

Implementing a smoking cessation intervention for people experiencing homelessness: participants’ preferences, feedback, and satisfaction with the ‘power to quit’ program

PONE-D-21-22291R2

Dear Dr. Odukoya,

We’re pleased to inform you that your manuscript has been judged scientifically suitable for publication and will be formally accepted for publication once it meets all outstanding technical requirements.

Kind regards,

Jesse T. Kaye, PhD

Academic Editor

PLOS ONE

Additional Editor Comments (optional):

Thank you for this important contribution to the literature and addressing the comments and suggestions from previous reviews.

In table 3 there appears to be a few typos in the response options where 'somewhat satisfied' is listed twice in several sections where I believe it should be 'somewhat unsatisfied' first and 'somewhat satisfied' second (on either side of neutral). Please correct this during the copyediting process.
---

## [Editor Report · Acceptance letter]

24 May 2022

PONE-D-21-22291R2 

Implementing a smoking cessation intervention for people experiencing homelessness: participants’ preferences, feedback, and satisfaction   with the ‘power to quit’ program 

Dear Dr. Odukoya:

I'm pleased to inform you that your manuscript has been deemed suitable for publication in PLOS ONE. Congratulations! Your manuscript is now with our production department. 

Kind regards, 

on behalf of

Dr. Jesse T. Kaye 

Academic Editor

PLOS ONE